**Data Availability Statement:** All data is available archived in Supporting Information files.

**Funding:** This study was partially funded by a grant from the Ministry of Science and Higher Education of the Russian Federation (agreement #075-15-

# Thermostable chaperone-based polypeptide biosynthesis: Enfuvirtide model product quality and protocol-related impurities

Vladimir Zenin[1]*, Andrey Tsedilin[1], Maria Yurkova[1], Andrey Siniavin[2], Alexey Fedorov[1]

1 Bach Institute of Biochemistry, Research Center of Biotechnology of the Russian Academy of Sciences, Moscow, Russian Federation, 2 Ivanovsky Institute of Virology, N.F. Gamaleya National Research Center for Epidemiology and Microbiology, Ministry of Health of the Russian Federation, Moscow, Russian Federation

* zenanmail@icloud.com

## Abstract

Large peptide biosynthesis is a valuable alternative to conventional chemical synthesis. Enfuvirtide, the largest therapeutic peptide used in HIV infection treatment, was synthesized in our thermostable chaperone-based peptide biosynthesis system and evaluated for peptide quality as well as the profile of process-related impurities. Host cell proteins (HCPs) and BrCN cleavage-modified peptides were evaluated by LC-MS in intermediate. Cleavage modifications during the reaction were assessed after LC-MS maps were aligned by simple in-house algorithm and formylation/oxidation levels were estimated. Circular dichroism spectra of the obtained enfuvirtide were compared to the those of the chemically- synthesized standard product. Final-product endotoxin and HCPs content were assessed resulting 1.06 EU/mg and 5.58 ppm respectively. Peptide therapeutic activity was measured using the MT-4 cells HIV infection-inhibition model. The biosynthetic peptide $IC_{50}$ was 0.0453 µM while the standard one had 0.0180 µM. Non-acylated C-terminus was proposed as a cause of $IC_{50}$ and CD spectra difference. Otherwise, the peptide has met all the requirements of the original chemically synthesized enfuvirtide in the cell-culture and *in vivo* experiments.

## Introduction

Peptides are a rapidly growing therapeutic group with more than 80 different peptides on the market [1]. We have developed a straightforward biosynthesis process for conventional ribosome- synthesized peptides. In previous articles we showed that it can be a useful and robust tool for peptide synthesis [2–4]. To prove its usability for pharmaceutic development of active peptide substances there are some points that must be examined.

The usual process of active substance development includes target validation, compound screening, secondary assays and *in vivo* analysis [5].

Starting from the screening, the method of compound synthesis matters. We expect that peptide biosynthesis is fully compatible with phage display technology [6]: its product is an

2021-1071). Andrey Siniavin performed "in vitro activity" part of the study without specific funding. The funders had no role in study design, data collection and analysis, decision to publish, or preparation of the manuscript.

**Competing interests:** The authors have declared that no competing interests exists.

encoded linear amino acid sequence. So, theoretically, most of the obtained peptide sequences can be readily transferred for biosynthesis in our system.

The next two steps usually include cell culture experiments and *in vivo* analysis. Therefore, the final product should meet the following criteria for material for cell culture/*in vivo* experiment:

- pH/osmolarity compatible

- sterile/viral free

- apyrogenic

- free of toxic impurities

The pH/osmolarity can be corrected, and particle absence can be ensured during sample preparation, while sterility, the presence of endotoxins and toxic impurity content depend on the material's origin and purification process.

Protein and peptide drugs are thermolabile, and the most common sterilization technique for such samples is filtration. Usually, it's enough to prevent microbial growth in a sample, and one can assume that the sample is sterile. However, it should be considered otherwise. Mycoplasmas are among the smallest self-replicating organisms [7]. Viruses are even smaller and some viruses are small enough even on HPLC column pore size scale, for example porcine circovirus type 1 is about 17 nm in diameter [8]. Therefore, sterilization by filtration is effective only for mycoplasma and virus-free samples.

Similar to chemical synthesis, the *E. coli* biosynthesis is not a source of mycoplasma or eukaryotic viruses. It is debatable, what process is easier to perform in an aseptic manner–chemical synthesis or biosynthesis with their purification steps.

The process impurities profiles of chemically synthesized and biotechnologically derived peptides are different. Chemically synthesized peptides potentially contain products of fragmentation, deletion, β-elimination, racemization, not fully deprotected products [9].

In this study, we can expect the following host cell soluble impurities co-purified with our protein: host proteins, host cell DNA, lipids (only endotoxins as a special case otf lipids will be regarded), and cleavage-induced impurities.

The host cell proteins (HCPs) level is regulatory limited to 100 ng/mg. Also, HCPs are a heterogeneous group. The most dangerous agents among them are physiologically or enzymatically active proteins, such as bacterial flagellins [10] or proteases [11]. We had the opportunity to explore the co-purifying fraction of HCPs in intermediate product by HPLC-MS to access potential biosynthesis system-related hazards directly.

The practical significance of *E. coli* residual DNA limit in pharmaceuticals is unclear [12, 13]. Immunogenicity concerns are raised [10]. However, practical study design usually does not exclude such prominent factors [14] as aggregates, endotoxins and HCPs. We decided to skip residual DNA test as the less concerning impurity for *in vitro/in vivo* tests.

Unlike the chemical synthesis, the *E. coli* is a source of endotoxin and its content in peptide must be justified prior any cell culture or *in vivo* usage.

Endotoxins are lipopolysaccharides (LPS) of gram-negative bacteria and strong activators of innate immunity [15]. Endotoxins cause intense system reactions when administered parenterally with a drug formulation and can affect cell cultures [16], so endotoxin levels are regulatory limited.

The cyanogen bromide cleavage process clearly produces protein modification. Also, this method limits the amino acid composition and puts the homoserine lactone on the C-termini of the released peptide. This mode of cleavage was chosen because of the small specific site

footprint, cheapness, and robustness. This method is not obligatory, the mode of subsequent treatment can be adapted for an individual proposed peptide with the construct adapted accordingly. The current hydrolysis protocol is expected to be the source of formylated and oxidized forms and C-terminal homoserine/homoserine lactone [17].

The crucial point of any active substance synthesis is the substance activity itself. The enfuvirtide acts as an antiretroviral fusion inhibition drug. MT-4 cells were used as an established *in vitro* model for anti-HIV drugs [18].

To investigate the difference between the enfuvirtide standard and our sample activities, the CD spectrum of the standard was obtained and compared with the sample CD spectrum.

## Materials and methods

Modified GroEL-Enfuvirtide fusion was produced as described previously [3] The protein with the designed sequence (S1 File) was expressed in *E. coli* cell culture in soluble form, purified by heat-induced host protein denaturation and IEX chromatography, desalted, lyophilized, and stored at -20˚C.

### Chemicals

Enfuvirtide Primary Reference Standard (T-20, Fuzeon) Roche Diagnostics GmbH.

Cyanogen bromide reagent grade from Sigma, USA, buffer components and SDS-PAGE reagents "for biochemistry" grade, by Amresco, USA; Pierce Unstained Protein MW Marker by Thermo Fisher Scientific, USA; formic acid for biochemistry by AppliChem, USA and MS-grade solvents by Merck, Germany were used.

### The hydrolysis process

Modified GroEL-Enfuvirtide fusion was hydrolyzed at 5 mg/mL concentration in 63.6% formic acid with 0.46 M cyanogen bromide and 4.6% acetonitrile. The reaction mixture was aliquoted and the process was performed at 20˚C in a dark place. Each hour, corresponding aliquots were quenched by x10 water dilution and immediate freezing in liquid nitrogen. Aliquots were dried in Alpha 3, 4 LSCbasic (Martin Christ, Germany) freeze-dryer at 0.05 mBar pressure and -110˚C condenser temperature.

Alternative protocols test. Gua 6M with 0.1 M HCl, 0.46 M cyanogen bromide and 4.6% acetonitrile and TFA instead of FA in original protocol were compared with original protocolat points 1 h and overnight (16 h). Initial protein concentration was equal and SDS-PAGE samples were prepared with adjustment by initial substrate concentration.

### Process-related impurities profiling

HPLC-MS/MS analysis was performed as described earlier [3]. Impact II QqTOF high-resolution mass-spectrometer (Bruker Daltonik, Germany) equipped with Elute UHPLC (Bruker Daltonik, Germany) and Intensity Solo 1.8 C18-2 2.1*100 mm 1.8 μm 90 Å reverse-phase column (Bruker Daltonik, Germany) was used. Chromatography was performed in an acetonitrile gradient with 0.1% (v/v) formic acid as additive.

### HCPs search

Tryptic digest was performed in solution with Trypsin Gold (Promega, USA) according to its user manual. Purified fusion protein was studied as an intermediate and purified peptide as the final product.

Modifications search samples were prepared as previously described for HPLC-MS/MS analysis.

## Peptide matching script

Resulting "expected and found" and "unidentified" peptide lists for all 1–7 h samples were combined. The reference peaks were chosen: the main product and its dimer, 3 different formylated and one oxidized form of the main product. Their molecular weights (MWs) and retention times (Rts) were normalized to theoretical MWs and mean Rts. Re-calibration between samples was performed using the mean of normalized MW and Rt of every sample. Normalized MW and Rt after recalibration were used for SD estimation. MW and Rt were scaled by estimated SD. The pairwise euclidean distance in MW and Rt 2D space was calculated and 3SD was used as a threshold for peak matching. The resulting adjacent matrix was converted into a list of connected graphs. Graphs were additionally matched by MW due to different Rt of the same modification on different AAs. The "expected" part contained multiple proposed peak options as individual peaks with different theoretical MW for every proteoform, the nearest by MW was chosen. In the whole graph, peptides were attributed by nearest to mean graph MW proposed peak option from "expected". If there were no proposed peak at a distance of 3SD, automatic attribution by BioCompass was considered incorrect.

The modification intensity of the sample was calculated as the sum of the modified peak intensity multiplied by its modification count. (Like: double formyl group gives double formyl intensity).

## Circular dichroism comparison

Spectra were collected by Chirascan circular dichroism spectrometer (Applied Photophysics, UK) as described previously. The sample was prepared at 2.5 mg/mL concentration in a 30 mM sodium phosphate buffer.

To exclude the impact of formic acid treatment on the secondary structure, following control sample was prepared: the reference sample was dissolved in a hydrolysis mixture, quenched and freeze-dried, dissolved in mobile phase 0.1% (v/v) formic acid in 30% acetonitrile, and dried overnight on rotary vacuum concentrator RVC 2–25 CDplus (Martin Christ, Germany) at 30˚C.

### *In vitro* activity

**Cells.** The human T lymphoblastoid cell line MT-4 (HTLV-1 transformed) was cultured in RPMI 1640 complete medium (Gibco, USA) supplemented with 10% fetal bovine serum (FBS; Sigma, USA), 1% GlutaMAX (Gibco, USA) and 1% antibiotics penicillin-streptomycin (Gibco, USA). Cells were passaged three times a week and cultured at a density of $<0.5{\times}106$ cells/mL.

**Viruses.** The viral stock of the laboratory strain HIV-1 IIIB (NIH AIDS Reagent Program) was obtained during acute infection of MT-4 cells. The virus was stored in aliquots at -80˚C.

**Antiviral assay.** The antiviral activity of the compounds against the HIV-1 IIIB strain in MT-4 cells was evaluated using a tetrazolium-based colorimetric assay as described recently [19]. Briefly, this method is based on HIV-induced cytopathic effect (CPE) in MT-4 cells 5 days post infection. The antiviral effects of the test compounds were directly correlated with the inhibition of virus-induced CPE by measuring cell viability using the MTT assay. MT-4 cells (6 x 105 cells/ml) were infected with the IIIB virus strain at 100 $CCID_{50}$ (50% cell culture infectious dose) in the presence of different compound dilutions. Protection against HIV-induced CPE was assessed using the MTT assay 5 days post-challenge.

IC$_{50}$ and its 95% confidence interval (CI) estimation were performed using GraphPad prism software (8.0.1 Build 244) and scipy 1.7.3 and uncertainties 3.1.7 python packages. GraphPad function Y = Bottom + (Top-Bottom)/(1+10^((LogIC50-X)*HillSlope)) was used for curve fitting with scipy.optimize.curve_fit. 83% CI was calculated as 1.39 of standard deviation and used of estimation of 95% probability for standard and sample CIs overlap [20].

### Other methods

SDS-PAGE in tris-glycine buffer and 10–20% gradient gel was performed as previously described [3].

Endotoxin assay was performed according to harmonized EP/USP "Bacterial endotoxin" monograph. The maximal valid dilution (MVD) calculation was performed with the assumption of 2 mg per kg subcutaneous administration with resulting 2,5 EU/mg. Controls were set accordingly "Confirmation of Labeled Lysate Sensitivity" and "Test for Interfering Factors".

Kinetic turbidimetric method was used with a ½ MVD sample concentration.

HCPs assay kit «E.coli Host Cell Proteins» (Cygnus Tecnologies, USA) was used according to its manual.

### Results

Comparison between guanidine with hydrochloric acid, formic acid and trifluoroacetic acid as media for cleavage showed significant sample loss for guanidine sample and fusion protein fragmentation for trifluoroacetic acid (Fig 1).

The SDS-PAGE of analytical 0–5 h hydrolysis in formic acid was previously published with no visible improvement in process yield after the first hour [3].

HCPs found by LC-MS in the intermediate GroEL-Enfuvirtide product before cyanogen bromide hydrolysis and RP-HPLC purification.

All potentially detected HCPs are listed in S1 File. Host proteins detected with high probability (with a MASCOT score above 90) [21] are presented in Table 1 below.

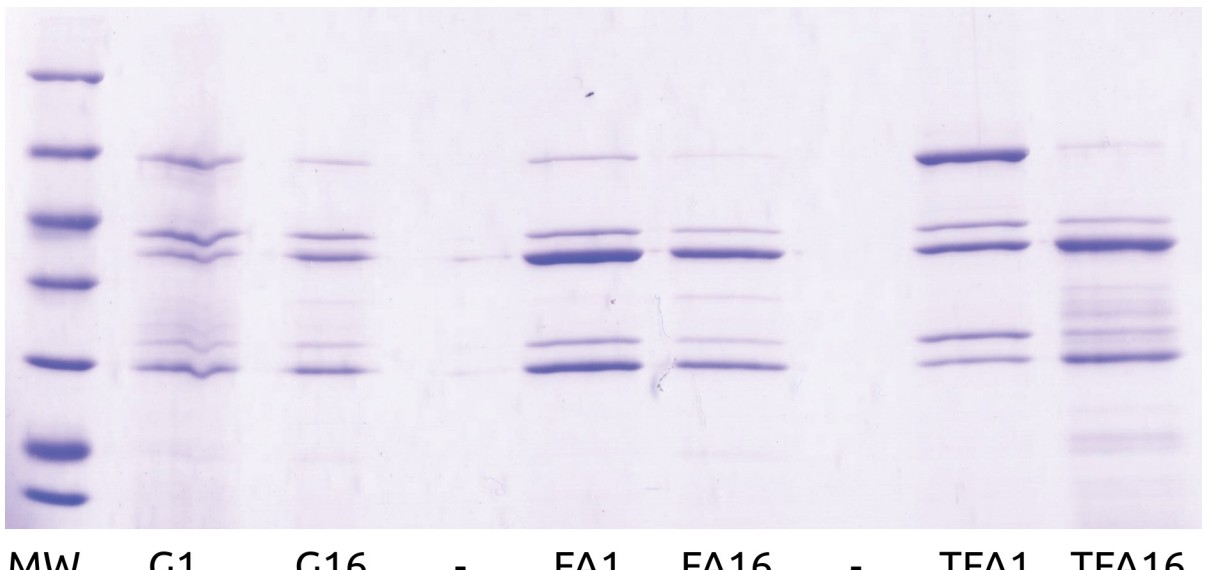

**Fig 1. SDS-PAGE of cyanogen bromide hydrolysis of fusion protein in different medium.** MW—molecular weight ladder 116, 66.2, 45, 35, 25, 18.4, 14.4 kDa, G—guanidine with hydrochloric acid, FA—formic acid, TFA—trifluoroacetic acid, 1 and 16 is hours since reaction start.

**Table 1. Host cell proteins detected in GroEL-Enfuvirtide intermediate product with high probability.**

| Score | SC [%] | #Peptides | pI | MW [kDa] | Protein |
|---|---|---|---|---|---|
| 811.1 | 42 | 15 | 5.2 | 77.5 | Elongation factor G |
| 762.3 | 39.4 | 13 | 4.8 | 57.3 | 60 kDa chaperonin |
| 484.7 | 33.8 | 7 | 5.3 | 45.6 | Enolase |
| 405.1 | 13.2 | 5 | 4.8 | 69.1 | Chaperone DnaK |
| 313.7 | 16.6 | 4 | 4.8 | 47.8 | Trigger factor |
| 310.9 | 19.5 | 9 | 5.3 | 43.3 | Elongation factor Tu 1 |
| 277.1 | 49 | 8 | 5.4 | 25.9 | Purine nucleoside phosphorylase DeoD-type |
| 245.4 | 11.1 | 5 | 4.9 | 61.1 | 30S ribosomal protein S1 |
| 245.3 | 19 | 5 | 4.6 | 12.3 | 50S ribosomal protein L7/L12 |
| 239.1 | 45 | 6 | 5.2 | 15.2 | 30S ribosomal protein S6 |
| 209.4 | 12.7 | 6 | 5.1 | 77.1 | Polyribonucleotide nucleotidyltransferase |
| 193.4 | 23.7 | 5 | 5.2 | 30.4 | Elongation factor Ts |
| 160.9 | 24 | 4 | 5.3 | 19.4 | S-ribosylhomocysteine lyase |
| 157.1 | 35.5 | 4 | 4.7 | 21.8 | Protein GrpE |
| 152.4 | 19.9 | 2 | 4.9 | 20.8 | FKBP-type peptidyl-prolyl cis-trans isomerase |
| 121 | 15.2 | 2 | 4.5 | 21 | Fe/S biogenesis protein NfuA |
| 113.4 | 42.3 | 1 | 4 | 8.6 | Acyl carrier protein |
| 104.8 | 3.8 | 2 | 5.1 | 66.1 | Dihydrolipoyllysine-residue acetyltransferase component of pyruvate dehydrogenase complex |
| 103.6 | 12.9 | 2 | 5.4 | 19 | Single-stranded DNA-binding protein |
| 102.2 | 35.8 | 3 | 5.4 | 15.5 | DNA-binding protein H-NS |
| 97.6 | 8.2 | 3 | 4.8 | 63.5 | Phosphoenolpyruvate-protein phosphotransferase |
| 93.4 | 12.5 | 3 | 5.8 | 43.3 | Acetate kinase |
| 92.1 | 34 | 3 | 5.1 | 16.1 | Universal stress protein A |

HCP level in final product was 5.58 ppm and bacterial endotoxin level less than 1.06 EU/ mg was measured.

After 1–7 h of BrCN hydrolysis peptide concentration and modifications level were assessed by HPLC-MS with subsequent peptide search and peak matching.

The peak matching process revealed an increased distribution density of pairwise distances in the region 0–3 SD (Fig 2).

Formylation proceeds with new multi-formylated forms appearing after second hour and S31 is the most vulnerable AA. Oxidation and hydrolysis of C-terminal homoserine lactone to homoserine are much slower (Fig 3A). The main product concentration declined over time (Fig 3B).

CD spectra of the reference peptide and synthesized sample with additional C-terminal methionine differ greatly (Fig 4). Maximal similarity between sample and standard is observed between CD spectra of sample at 65°C and standard at 20°C.

The *in vitro* HIV-1 infection model revealed 2.5-fold lower mean IC50 for bio-synthetic modified enfuvirtide 0.0453 μM [95% CI 0.0316–0.0650] while the standard had 0.0180 μM [95% CI 0.0090–0.0359]. Infection inhibition curves compared at Fig 5.

Circular dichroism data and visualization script, mass-spectrometry BioCompass search results and previously described script, NIR results, processing and visualization script, HCP and IC50 test protocols are included in S2 File.

## Discussion

Most of the listed HCPs are abundant *E. coli* cell proteins and well-known downstream impurities.

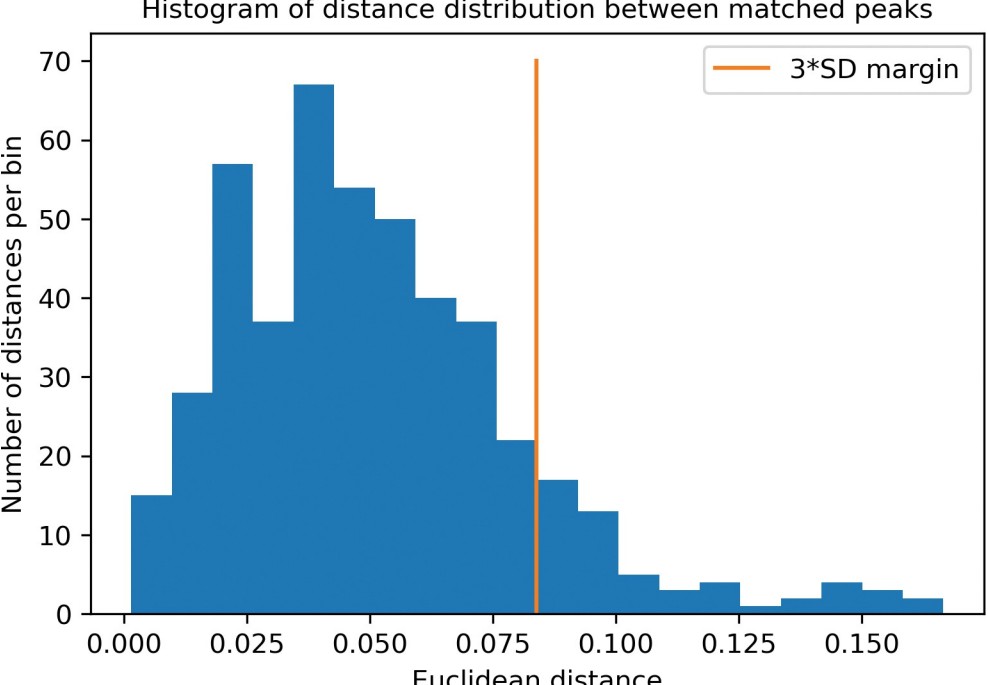

**Fig 2. Histogram of pairwise Euclidean distance distribution during peak matching in 2D (MW and Rt) space after SD scaling.**

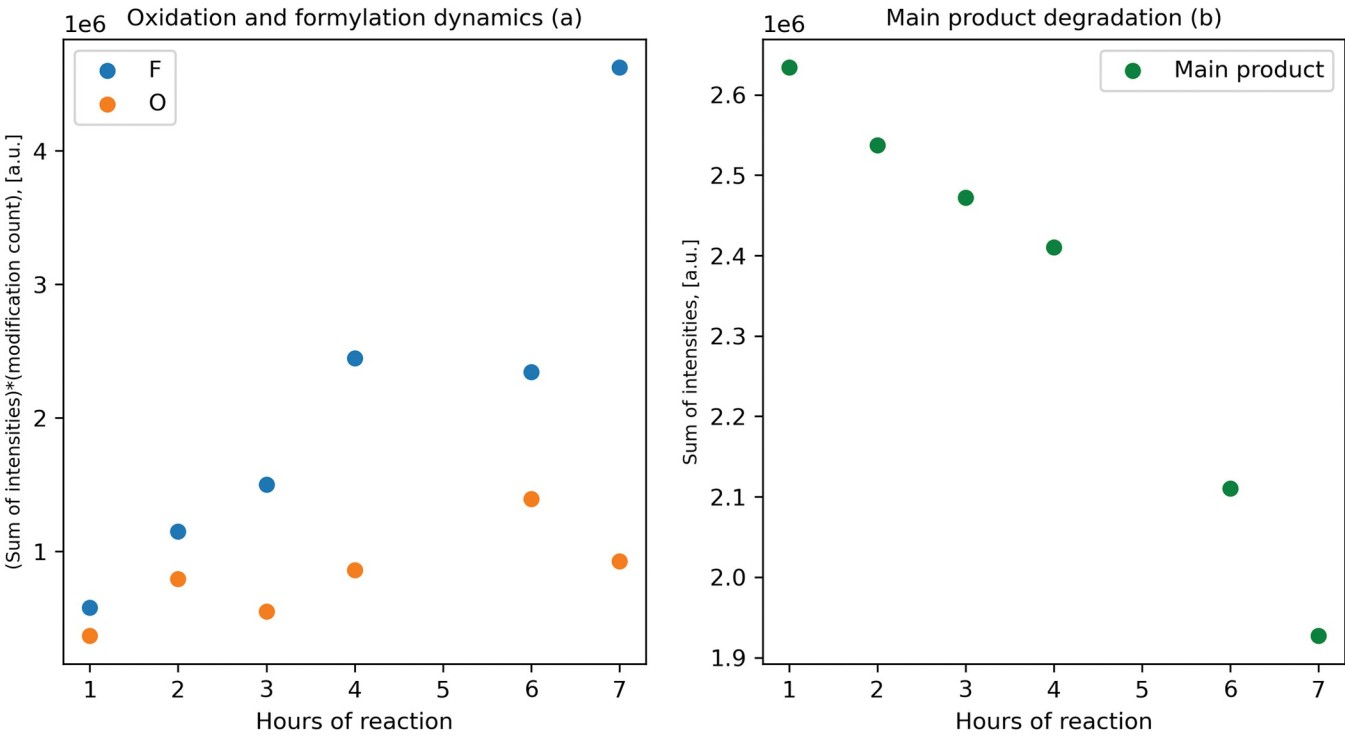

**Fig 3.** a–Product formylation (orange) and oxidation (green) rate, b–main product abundance in samples.

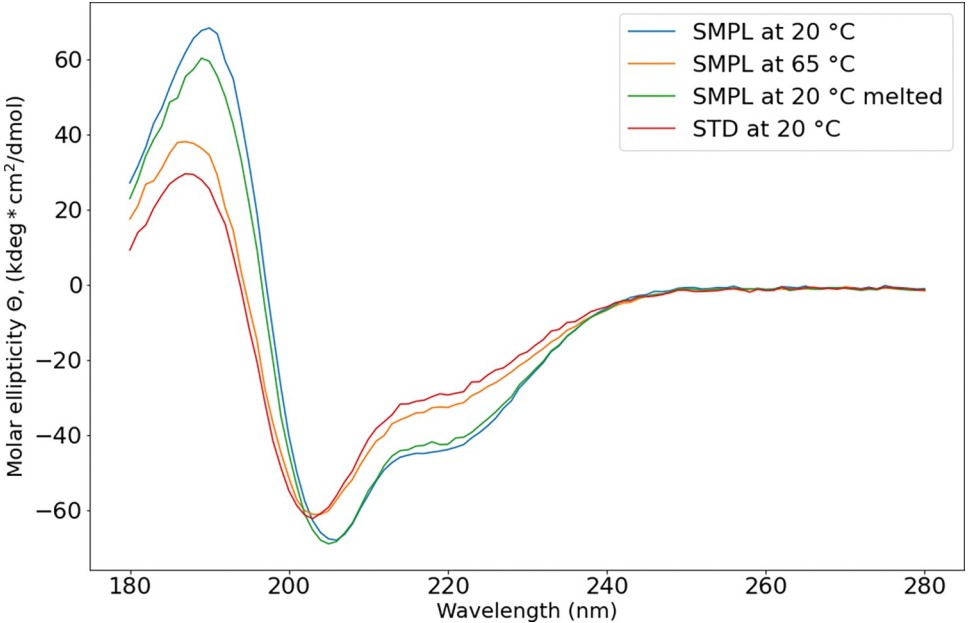

**Fig 4. Circular dichroism spectra of biosynthesized enfuvirtide (SMPL) and reference standard enfuvirtide (STD) at different temperature.** "Melted" stands for heating to 90˚C and cooling to 20˚C at 1˚C /min in device cuvette.

Among the established high-risk HCPs [22] are the following:

- Enolase (drug modification attributed, also the reference publication has other protein as a root of the problem [23])

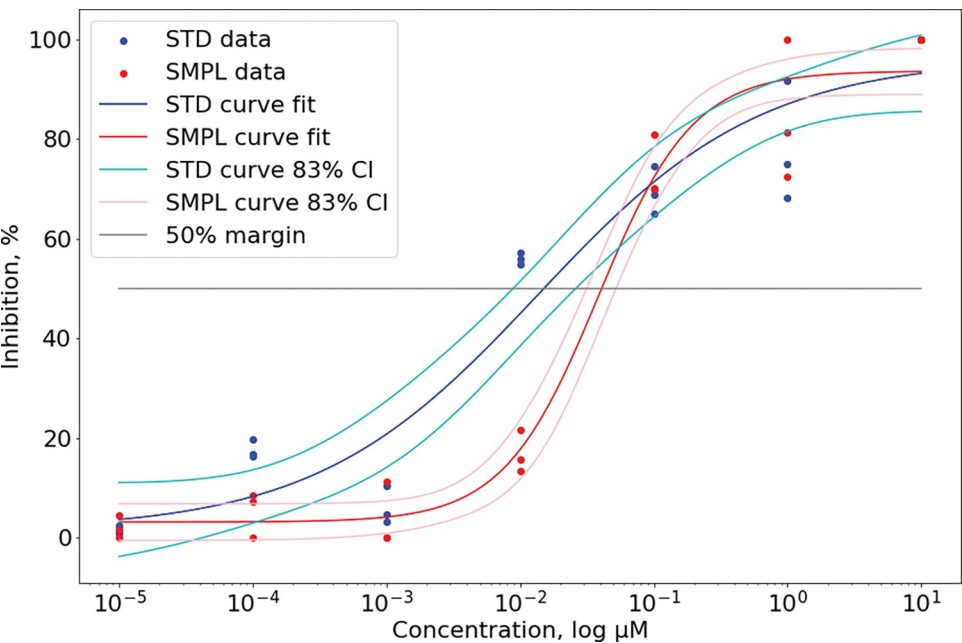

**Fig 5. HIV-1 infection inhibition on MT-4 cells for biosynthesized enfuvirtide (SMPL) and reference standard enfuvirtide (STD).** 83%CI was used as threshold for statistically significant difference (p = 0.05) on two CI intersection.

- FKBP-type peptidyl-prolyl cis-trans isomerase SlyD (Peptidyl-prolyl cis-trans isomerase A) has mentioned as an aggregation factor.

- Pyruvate kinase I (CHO PK immunogenicity [24] by IEDB and EpiMatrix db's)

- DnaK chaperone has potential to enhance immunogenicity while binding to aggregated active ingridient protein [25].

Overall, the absence of proteases is a good feature. However, peptidyl-prolyl cis-trans isomerase with its ability to change the proper conformation of some proline-containing peptides and DnaK with its immunogenicity issues are significant factors to consider during subsequent hydrolysis and purification process [22–25]. In this study most of proteins will be cleaved on M sites, so no enzymatic activity is expected. BrCN cleavage usually yields large peptides and immunogenicity is still a concern for non-purified product.

Endotoxin concentration less than 1.06 EU/mg was measured. It is lower, than 2.5 EU/mg limit for high-dosage (90 mg per dose) enfuvirtide drug.

Endotoxins interact with proteins and are readily co-purified during downstream process [26]. These interactions are expected to have mixed electrostatic and hydrophobic nature with Ca2+ ions involved [27]. Relying on the final RP-HPLC step, we did not pursuit the LSP removal during intermediate GroEL-enfuvirtide purification. So, if peptide biosynthesis system re-engineering revises the final RP-HPLC step–some LPS removal measures should be added.

The ordinary ion-exchange chromatography step we used cannot provide significant LPS removal [28] due to its hydrophobic interactions with purified protein.

Apparently, detergent can disrupt hydrophobic interactions between protein and LPS while separation is performed in different mode–ion exchange like in this study, or any other compatible generally. We suppose including an additional step of non-ionic detergent washing [29] in that case.

Protein formylation in formic acid solutions up to 0,1% dilution is well-known [30]. Nonetheless, formic acid usage is practically unavoidable in analysis and, sometimes, in preparative processes.

In this study, for cyanogen bromide cleavage, we had to stay on the formic acid option instead of guanidine or trifluoroacetic acid. We assume that formation of new low molecular bands with TFA or "shadow" with guanidine and HCl on SDS-PAGE gel can be result of protein fragmentation by non-specific hydrolysis as well as low recovery of hydrolysis products.

We re-evaluated formylated product content and added new non-matched automatically peaks.

We used our own framework for peptide matching with the results of Bruker software.

The raw data were processed by BioCompass successfully, with a resulting "expected" and "unexpected" peptide lists. Some peaks were listed in "expected" multiple times with different theoretical proteoforms proposed. Also, BioCompass software has a peculiarity: after the first discovery of a proteoform on the LC profile, the software ceases the search of that proteoform. Therefore, we must choose the proper proteoform and validate the assignment, to search its isomers on the LC profile and match all peptides between the samples.

Specifically, we encountered a task, known as LC-MS feature map alignment.

There are many LC-MS alignment algorithms for raw map alignment or feature map alignment with various software implementations [31, 32]. The Jupyter Notebook is the de facto standard for data exploration [33]. The interactive nature of Jupyter and the flexibility of the Python language offer a great advantage for data processing and presentation. There are some sophisticated Jupyter-ready pipelines: AlphaPept [34], BioDendro [35], TidyMS [36], designed

for raw data processing. However, their entry threshold is steep. So we created a minimalistic tool for processed data, meeting our needs.

The resulting experimental peptide mass (or Rt) fluctuates, depending on several factors. Some of these are unique for a given peptide peak and some of them–for the whole LC run. We have been able to compensate run-dependent variability, resulting in 1.5x SD reduction–in fact, we recalibrated Rt and mass measurement for every sample, using hand-picked set of different reference peaks.

We evaluated peak-dependent dispersion after recalibration. The peak probability distribution in this study was close to normal so we used the empirical rule to estimate the statistically probable difference between the same peak in different samples after recalibration. After SD scaling, we used Euclidean distance as the symmetric metric of the Rt AND mass difference. The histogram of distance distribution shows that run-dependent variability was compensated sub-optimally with systematic error. Mostly, this error is provided by mass calibration shift on dimeric enfuvirtide high-mass reference peak. This effect can be avoided by reference peak sample size increase and usage of linear function for calibration correction instead of a single coefficient.

Our algorithm is close to the gross-alignment in XAlign [37], but we use mean Rt instead of median Rt for Rt recalibration and added mass recalibration by theoretical mass of manually validated assigned peptides. The empirical rule in our method plays the same role as two-dimensional Kolmgorov–Smirnov (K–S) test in XAlign.

After initial peak matching mass measurement was refined and peptide isomers were matched by mass only.

As we can see, anybody who uses the standard overnight BrCN protocol has his product heavily modified. Modification rates on the 1–7 h interval are close to linear, so hydrolysis time optimization is effective way to facilitate further purification and increase yield. Hydrolysis time optimization on model peptide is not justified: susceptibility to BrCN hydrolysis differs for different M-X pairs and, presumably for different sequences [38].

The formyl modification occurs naturally–N-formyl methionine peptides are involved in chemotaxis and inflammation [39] and lysine formylation of histones has epigenetical function [40]. Definitely, formylation impacts peptide charge and conformation, thus affecting the activity and should be avoided.

Formylation is temperature-dependent [41], like any other chemical reaction and so the BrCN cleavage, and there is no obvious point in low-temperature hydrolysis.

Oxidation depends on BrCN reagent quality [17], in our study we used the properly stored, but not the fresh opened one.

Circular dichroism data show that reference enfuvirtide has less α-helical structure, than biosynthesized. Possible explanation may include involvement of free N-terminus in secondary structure formation (DichroWeb structures approximations in S1 File). Additionally, structural differences were registered by NIR spectra [42] (method in S1 File).

The revealed activity difference between standard and bio-synthesized enfuvirtide has its roots in their chemical or/and conformational nonidentity.

Firstly, standard enfuvirtide has its N- and C-termini endcapped by acetylation and amidation. The bio-synthesized one has a free N-terminus and an additional homoserine lactone on C-terminus. The latter has much lower polarity than the free C-terminus and in that way, it is comparable with amidated C-terminus. Also, they differ sterically. Free tyrosine N-terminus is basic and it will be charged under physiological conditions.

Wild and colleagues presented α-helical peptides as anti-HIV agents and suggested endcapping because such peptides are derived from the longest chain "The N terminus of the peptide was acetylated and the C terminus amidated to reduce unnatural charge effects at those

positions" [43]. Therefore, most ever published antiretroviral α-helical peptides have N- and C- termini blocked [44–47].

The enfuvirtide mechanism of action involves gp41 fusion intermediate stabilization by 6 strand formation with 3 α-helical enfuvirtide peptide units and 3 gp41 protein strands [44, 48, 49].

Zhang and colleagues described the model complex of gp41-derived peptide N39 and enfuvirtide: "...the first residue Tyr-127 not only interacts with His-53 on N39 by a hydrogen bond but also contacts with Leu-54 by hydrophobic force..." [49, p3] and "...Trp-155 and Ala-156 mediate hydrophobic contacts with Leu-26, which locate at the TRM (tryptophan-rich motif) of T20 and the FPPR (fusion peptide proximal region) of N39, respectively..." [49, p3]. Also, there is a data, that C-terminus of enfuvirtide is interacting with the cell membrane while blocking cell and viral membrane fusion and hydrophobic addition such as octyl- residue on C-terminus will boost the peptide activity [50]. So, the absence of charged N and C-termini is crucial for maximal affinity.

Therefore, the activity difference is most likely explained by the charged non-acetylated N-terminus of biosynthesized sample and less likely by its sterically different homoserine lactone on C-terminus.

Further investigation of activity difference may include biosynthesized enfuvirtide selective N-terminus acetylation. However, this procedure is a state-of-the-art of selective modification [51] and it's usage on model peptide is not justified. *In silico* methods for estimation of non-acetylated N-terminus and C-terminal homoserine lactone contribution on activity difference can be of some help. Yet, it will be not ground truth and such estimation will not assess the general applicability of present peptide biosynthesis method.

## Conclusion

The proposed biosynthesis method yields a model peptide product with a cell-culture/*in vivo*-ready purity. Our method has several advantages: no chemical synthesis reagents and consumables needed, less toxic solvents used, can be introduced in common biotech lab, *E. coli* strain can be stored at -80˚C for next fermentations. It can be used for ribosomal biosynthesis of large peptides without methionine amidst the sequence **as is**, and with methionine after cleavage site and cleavage protocol revision.

Enfuvirtide model shows nuances in required modification introduction (like N-terminus acylation in our case) for biotechnologically-derived peptides. Necessary modifications, if any, should be considered at the outset.

Overall, the proposed biosynthesis method is suitable for a significant proportion of large (up to 50 AA) peptide biosynthesis cases. The sufficient purification step is included so method can be applied as presented.

There is a room for further upgrades of the method. The future development tracks are outlined: the native conditions His-tag introduction is underway; the system ability to accommodate peptide as large as 100 AA and alternative cleavage protocols are discussed.

## Supporting information

**S1 File.**
(DOCX)

**S2 File.**
(ZIP)

**S1 Graphical abstract.**
(TIFF)

## Acknowledgments

The measurement were carried out on the equipment of the Shared-Access Equipment Centre "Industrial Biotechnology" of Federal Research Center "Fundamentals of Biotechnology" Russian Academy of Sciences.

The following reagent was obtained through the NIH HIV Reagent Program, Division of AIDS, NIAID, NIH: MT-4 Cells, ARP-120, contributed by Dr. Douglas Richman.

Enfuvirtide Primary Reference Standard (T-20, Fuzeon) was generously provided by Roche Diagnostics GmbH.

## Author Contributions

**Conceptualization:** Vladimir Zenin, Maria Yurkova, Alexey Fedorov.

**Data curation:** Vladimir Zenin.

**Formal analysis:** Vladimir Zenin.

**Funding acquisition:** Alexey Fedorov.

**Methodology:** Vladimir Zenin, Andrey Tsedilin, Maria Yurkova, Andrey Siniavin.

**Supervision:** Maria Yurkova, Alexey Fedorov.

**Writing – original draft:** Vladimir Zenin, Andrey Tsedilin, Andrey Siniavin.

**Writing – review & editing:** Andrey Tsedilin, Maria Yurkova, Andrey Siniavin, Alexey Fedorov.

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
