## [Decision Letter · Decision Letter 0]

6 Feb 2023

PONE-D-22-30362Thermostable chaperone-based polypeptide biosynthesis: enfuvirtide model product quality and protocol-related impuritiesPLOS ONE

Dear Dr. Zenin,

Thank you for submitting your manuscript to PLOS ONE. After careful consideration, we feel that it has merit but does not fully meet PLOS ONE’s publication criteria as it currently stands. Therefore, we invite you to submit a revised version of the manuscript that addresses the points raised during the review process.

The manuscript has been evaluated by two reviewers, and their comments are available below. The reviewers have raised a number of concerns that need attention.

In particular, due to technical considerations, they have suggested that some experiments require repeating (protein quantification). Furthermore, the reviewers have suggested additional data can strengthen the manuscript. And finally, the reviewers  have concerns around grammatical errors, and thus suggest copyediting for English language and grammar.

Could you please revise the manuscript to carefully address the concerns raised?

We look forward to receiving your revised manuscript.

Kind regards,

Richard Ali

Staff Editor

PLOS ONE

Journal Requirements:

- \\\\https://journals.lww.com/aidsonline/Fulltext/2019/01020/Structural_and_functional_characterization_of.1.aspx?

In your revision ensure you cite all your sources (including your own works), and quote or rephrase any duplicated text outside the methods section. Further consideration is dependent on these concerns being addressed.

Reviewers' comments:

Reviewer's Responses to Questions

**Comments to the Author**

1. Is the manuscript technically sound, and do the data support the conclusions?

Reviewer #1: Yes

Reviewer #2: Yes

2. Has the statistical analysis been performed appropriately and rigorously? 

Reviewer #1: I Don't Know

Reviewer #2: Yes

3. Have the authors made all data underlying the findings in their manuscript fully available?

Reviewer #1: Yes

Reviewer #2: Yes

4. Is the manuscript presented in an intelligible fashion and written in standard English?

Reviewer #1: No

Reviewer #2: No

5. Review Comments to the Author

Reviewer #1: The manuscript entitled “Thermostable chaperone-based polypeptide biosynthesis: enfuvirtide model product quality and protocol-related impurities” aimed Enfuvirtide used in HIV infection treatment was synthesized in a developed thermostable chaperone-based peptide biosynthesis system and evaluated for the peptide quality as well as process-related impurities profile. This study is original. There are many grammatical errors in the manuscript.

1. Some references were shown in square brackets, the others were shown in round brackets. For example, square brackets has been used in line 29, while round brackets has been used in line 35. The uniformity should be provided for the manuscript references

2. There are many grammatical errors in the manuscript. These are;

-Line 56 – “Thereby, sterilization by filtration is effective only for mycoplasma and virus-free samples”. Thereby should be replaced with the therefore,

-Line 58 – “Similarly to chemical synthesis……” should be changed as “Similar to chemical synthesis……”

-Line 116 – “Gua 6M with 0.1 M HCl, 0.46 M cyanogen bromide and 4.6% acetonitrile” is an incomplete sentence.

-Line 157 - (HPLC PROCESS??) has not been understood.

-Line 179 – “IC50 and its 95% confidence interval (CI) estimation was performed using Graphpad prism software (version?, mode)” was should replaced with were. Also, the part “version?” has not been understood.

-Line 181 - Graph pad should be written adjacent.

-Line 195 – in the last of the sentence “………… according to its manual.,” comma should be removed.

-Line 219 – the full stop should removed from the last of the sentence “…….. assessed by HPLC-MS.”

-Line 227 – In “The main prodict concentration declined over time”, prodict should be corrected as product.

-Line 247 – E. coli should be written italic.

-Line 257 to 260 – literature citation should be added.

-Reference list format should be checked because it contains some mistakes. For example, Reference 17 includes website address of the publisher and does not include page number. References should be according to the Plos One rules.

Reviewer #2: The article entitled “Thermostable chaperone-based polypeptide biosynthesis:enfuvirtide model product quality and protocol-related impurities” by Vladimir Zenin et al, reported about biosynthesis of Enfuvirtide, the therapeutic peptide used in HIV treatment. Biosynthetic methods definitely have few advantages over classical chemical synthesis. The author performed some interesting experiments in-support of their claim; result and discussion were also understandable. Manuscript overall scientific quality is good. However I have some serious doubts regarding their experiments which are listed below. So in my opinion manuscript in the present form the paper is not suitable for publication in Plos one

Major Points

1. It’s evident from Figure 1 that the loading of the peptides were not equal in each lane, for e.g. G1 and G16 lane peptides were loaded much lesser compared to FA1 and FA16, they can only quantify digestion if the proteins were loaded in same amount. So in my opinion authors should repeat the experiments with equal loading.

2. Authors reported “significant sample loss for guanidine sample and fusion protein fragmentation for trifluoroacetic acid” however author should provide necessary justification for their observation.

3. Author should also calculate disordered versus ordered structure in Enfuvirtide. It’s not clear to me why chemically manufactured Enfuvirtide is having more disorder? Secondly why their synthesized Enfuvirtide shows similar kind of CD spectra only in 20°C and 65°C why not in other temperatures, is there any disorder to order transition going on with increasing temperature ? In my opinion they should also study peptide conformation in near-UV spectra, this will provide some idea about peptide 3-D structure.

4. It’s not clear to me what is the final yield of Enfuvirtide? Is it comparable to chemical synthesis method or low?

5. Also authors should add a schematic diagram starting from peptide fusion strategy to its final purification

Minor Points

There are some serious grammatical error, indexing, punctuation and language issue throughout the manuscript so the manuscript needs serious English editing.

6. PLOS authors have the option to publish the peer review history of their article (what does this mean?). If published, this will include your full peer review and any attached files.

Reviewer #1: No

Reviewer #2: No

---

## [Author Response · Author response to Decision Letter 0]

9 May 2023

Answers on Reviewer's Questions

Comments to the Author

1. Is the manuscript technically sound, and do the data support the conclusions?

Reviewer #1: Yes

Reviewer #2: Yes

2. Has the statistical analysis been performed appropriately and rigorously?

Reviewer #1: I Don't Know

Reviewer #2: Yes

3. Have the authors made all data underlying the findings in their manuscript fully available?

Reviewer #1: Yes

Reviewer #2: Yes

4. Is the manuscript presented in an intelligible fashion and written in standard English?

Reviewer #1: No

Reviewer #2: No

5. Review Comments to the Author

Reviewer #1: The manuscript entitled “Thermostable chaperone-based polypeptide biosynthesis: enfuvirtide model product quality and protocol-related impurities” aimed Enfuvirtide used in HIV infection treatment was synthesized in a developed thermostable chaperone-based peptide biosynthesis system and evaluated for the peptide quality as well as process-related impurities profile. This study is original. There are many grammatical errors in the manuscript.

1. Some references were shown in square brackets, the others were shown in round brackets. For example, square brackets has been used in line 29, while round brackets has been used in line 35. The uniformity should be provided for the manuscript references

Check Zotero results for manuscript (glitched!)

2. There are many grammatical errors in the manuscript. These are;

-Line 56 – “Thereby, sterilization by filtration is effective only for mycoplasma and virus-free samples”. Thereby should be replaced with the therefore,

(line 48 now, replaced)

-Line 58 – “Similarly to chemical synthesis……” should be changed as “Similar to chemical synthesis……”

(line 50, changed)

-Line 116 – “Gua 6M with 0.1 M HCl, 0.46 M cyanogen bromide and 4.6% acetonitrile” is an incomplete sentence.

(corrected, lines 95-98)

-Line 157 - (HPLC PROCESS??) has not been understood.

(line 127-128 here was link to previous article protocol for me to remember. It is corrected now)

-Line 179 – “IC50 and its 95% confidence interval (CI) estimation was performed using Graphpad prism software (version?, mode)” was should replaced with were. Also, the part “version?” has not been understood.

(line 145 corrected and replaced by version and build)

-Line 181 - Graph pad should be written adjacent.

(line 146, corrected)

-Line 195 – in the last of the sentence “………… according to its manual.,” comma should be removed.

(line 156, removed)

-Line 219 – the full stop (dot) should removed from the last of the sentence “…….. assessed by HPLC-MS.”

(line 174, removed)

-Line 227 – In “The main prodict concentration declined over time”, prodict should be corrected as product.

(line 182, corrected)

-Line 247 – E. coli should be written italic.

line 199, corrected

-Line 257 to 260 – literature citation should be added.

line 207, I’ve summarized 4 lines properly cited just before, so I had not considered to repeat citation. Now this issue is corrected.

-Reference list format should be checked because it contains some mistakes. For example, Reference 17 includes website address of the publisher and does not include page number. References should be according to the Plos One rules.

(Had some technical issues, now they are resolved.)

Thank you for your comments and expertise, the notes in answer to your comments are in text for your convenience.

Reviewer #2: The article entitled “Thermostable chaperone-based polypeptide biosynthesis:enfuvirtide model product quality and protocol-related impurities” by Vladimir Zenin et al, reported about biosynthesis of Enfuvirtide, the therapeutic peptide used in HIV treatment. Biosynthetic methods definitely have few advantages over classical chemical synthesis. The author performed some interesting experiments in-support of their claim; result and discussion were also understandable. Manuscript overall scientific quality is good. However I have some serious doubts regarding their experiments which are listed below. So in my opinion manuscript in the present form the paper is not suitable for publication in Plos one

Major Points

Thank you for your expertise and engagement.

1. It’s evident from Figure 1 that the loading of the peptides were not equal in each lane, for e.g. G1 and G16 lane peptides were loaded much lesser compared to FA1 and FA16, they can only quantify digestion if the proteins were loaded in same amount. So in my opinion authors should repeat the experiments with equal loading.

Theoretically, quantities in samples are the same. In our opinion low recovery and «shadow» on background of G1 could be connected. . We think that the Figure 1 should be represented in such way to highlight low recovery in guanidine for the same initial amount of protein. The final concentration of hydrolysis products is the result we want to highlight. The “substrate”/”products” ratio in final solution is less relevant for process yield than “loaded substrate”/”product in final solution” ratio. Such representation of results is intentional. Sample preparation details were added in Material and Methods section to support and clarify our opinion.

2. Authors reported “significant sample loss for guanidine sample and fusion protein fragmentation for trifluoroacetic acid” however author should provide necessary justification for their observation.

Thank you for your comment. In our opinion, the emergence of new lower MW bands on SDS-PAGE of protein sample among with narrowing of main band is a clear sign of fragmentation. (Considering sample simultaneous dilution and contamination with protein material is unrealistic.) We think it is out of article scope to test that shadows and bands by MS methods to prove fragmentation further.

3. Author should also calculate disordered versus ordered structure in Enfuvirtide. It’s not clear to me why chemically manufactured Enfuvirtide is having more disorder? Secondly why their synthesized Enfuvirtide shows similar kind of CD spectra only in 20°C and 65°C why not in other temperatures, is there any disorder to order transition going on with increasing temperature ? In my opinion they should also study peptide conformation in near-UV spectra, this will provide some idea about peptide 3-D structure.

DichroWeb structures approximations were added, we changed our statement on disordered structure. Really, only α-helical content is changed substantially from sample to sample.

We checked 65°C because same temperature is used for lysate clarification by thermal denaturation of host proteins. While cleavage protocol can be changed to enzymatic, thermal denaturation step is a key feature of our system. So we checked the reversibility of structural processes during heating and revealed such a shift towards original structure.

In our opinion, the free dissolved peptide structure (and it is greatly differs from full α-helical in active form on HIV gp41) is less related to peptide activity. We concentrated on discussion of impact of chemical difference on peptide-HIV complex structure.

We thoroughly described chemical difference (lines 349-353) and discussed non-acylated C-terminus as the root of structural and antiviral activity difference between biosynthesised and chemically-derived enfuvirtide (lines 354-373). Our discussion was supported by several concordant opinions on C-terminal charge impact on enfuvirtide[1,2] and related peptides[3], and practical decisions[3–6].

We did not analyzed peptides structures in free dissolved form in depth because we used presented structures difference only as demonstration of peptides non-equivalence.

We performed NIR spectroscopy, yet our data has variability and our interpretation is limited – peptides are not identical structurally. In our opinion, structural studies are out of scope of present article.

1. Peisajovich SG, Gallo SA, Blumenthal R, Shai Y. C-terminal Octylation Rescues an Inactive T20 Mutant. J Biol Chem. 2003;278: 21012–21017. doi:10.1074/jbc.M212773200

2. Zhang X, Ding X, Zhu Y, Chong H, Cui S, He J, et al. Structural and functional characterization of HIV-1 cell fusion inhibitor T20. AIDS. 2019;33: 1–11. doi:10.1097/QAD.0000000000001979

3. Wild C, Oas T, McDanal C, Bolognesi D, Matthews T. A synthetic peptide inhibitor of human immunodeficiency virus replication: correlation between solution structure and viral inhibition. Proc Natl Acad Sci. 1992;89: 10537–10541. doi:10.1073/pnas.89.21.10537

4. Eckert DM, Kim PS. Design of potent inhibitors of HIV-1 entry from the gp41 N-peptide region. Proc Natl Acad Sci. 2001;98: 11187–11192. doi:10.1073/pnas.201392898

5. Su SB, Gong W, Gao J-L, Shen W-P, Grimm MC, Deng X, et al. T20/DP178, an Ectodomain Peptide of Human Immunodeficiency Virus Type 1 gp41, Is an Activator of Human Phagocyte N-Formyl Peptide Receptor. Blood. 1999;93: 3885–3892. doi:10.1182/blood.V93.11.3885

6. Rimsky LT, Shugars DC, Matthews TJ. Determinants of Human Immunodeficiency Virus Type 1 Resistance to gp41-Derived Inhibitory Peptides. J Virol. 1998;72: 986–993. doi:10.1128/JVI.72.2.986-993.1998

4. It’s not clear to me what is the final yield of Enfuvirtide? Is it comparable to chemical synthesis method or low?

The final yield was described in previous article and it is around 35% for lab hydrolysis and HPLC protocol

SPPS for enfuvirtide has 6–8% yield for linear process and up to 30% for SPPS and ligation with 7(!) intermediate purification steps for production-scale process(Bray, 2003).

Yes, it is comparable in theoretical to practical yield ratio, however the yield per liter of E. coli culture is around 3 mg now. It’s confusing, yet the culture was not even glucose-fed and has potential to increase density more than 10 times.

5. Also authors should add a schematic diagram starting from peptide fusion strategy to its final purification

The whole understanding of present article is not possible without previous one. The scheme was introduced in https://doi.org/10.1016/j.btre.2022.e00734 We did not consider to add the process scheme in present article. Yet, now we see how it helps to got the method in a glance, so we added the updated one, thank you. 

Minor Points

There are some serious grammatical error, indexing, punctuation and language issue throughout the manuscript so the manuscript needs serious English editing.

We done additional professional grammar check, thank you!

6. PLOS authors have the option to publish the peer review history of their article (what does this mean?). If published, this will include your full peer review and any attached files.

Do you want your identity to be public for this peer review? For information about this choice, including consent withdrawal, please see our Privacy Policy.

Reviewer #1: No

Reviewer #2: No

---

## [Editor Report · Decision Letter 1]

23 May 2023

Thermostable chaperone-based polypeptide biosynthesis: enfuvirtide model product quality and protocol-related impurities

PONE-D-22-30362R1

Dear Dr. Zenin,

We’re pleased to inform you that your manuscript has been judged scientifically suitable for publication and will be formally accepted for publication once it meets all outstanding technical requirements.

Kind regards,

Soumyananda Chakraborti

Guest Editor

PLOS ONE
---

## [Editor Report · Acceptance letter]

31 May 2023

PONE-D-22-30362R1 

Thermostable chaperone-based polypeptide biosynthesis: enfuvirtide model product quality and protocol-related impurities 

Dear Dr. Zenin:

I'm pleased to inform you that your manuscript has been deemed suitable for publication in PLOS ONE. Congratulations! Your manuscript is now with our production department. 

Kind regards, 

on behalf of

Dr. Soumyananda Chakraborti 

Guest Editor

PLOS ONE